# Icebreaker:
# Element-wise Efficient Information Acquisition with a Bayesian Deep Latent Gaussian Model

**Wenbo Gong**[1*], **Sebastian Tschiatschek**[2], **Richard E. Turner**[12],
**Sebastian Nowozin**[2†], **José Miguel Hernández-Lobato**[12], **Cheng Zhang**[2]

## Abstract

In this paper, we address the *ice-start* problem, i.e., the challenge of deploying machine learning models when only a little or no training data is initially available, and acquiring each feature element of data is associated with costs. This setting is representative of the real-world machine learning applications. For instance, in the health-care domain, obtaining every single measurement comes with a cost. We propose *Icebreaker*, a principled framework for element-wise training data acquisition. *Icebreaker* introduces a full *Bayesian Deep Latent Gaussian Model (BELGAM)* with a novel inference method, which combines recent advances in amortized inference and stochastic gradient MCMC to enable fast and accurate posterior inference. By utilizing BELGAM's ability to fully quantify model uncertainty, we also propose two information acquisition functions for *imputation* and *active prediction* problems. We demonstrate that BELGAM performs significantly better than previous variational autoencoder (VAE) based models, when the data set size is small, using both machine learning benchmarks and real-world recommender systems and health-care applications. Moreover, Icebreaker not only demonstrates improved performance compared to baselines, but it is also capable of achieving better test performance with less training data available.

## 1 Introduction

Acquiring information is costly in many real-world applications. For example, a medical doctor often needs to carry out a sequence of lab tests to make a correct diagnosis, where each of these tests is associated with a cost in terms of money, time, and health risks. To this end, an AI system should be able to suggest the information to be acquired in the form of "one measurement (feature) at a time" for accurate predictions (diagnosis) of any new user. Recently, test-time active prediction methods, such as EDDI (Efficient Dynamic Discovery of high-value Inference) [28], provide a solution for such a problem when there is a sufficient amount of training data. Unfortunately, in many scenarios, training data can also be challenging and costly to obtain. For example, new data needs to be collected by taking measurements of currently hospitalized patients with their consent. Ideally, we would like to deploy an AI system, such as EDDI, when no or only limited training data is available. We call this problem the *ice-start* problem.

The key to address the ice-start problem is to have a scalable model that *knows what it does not know*, namely to quantify the epistemic uncertainty. This knowledge can be used to guide the acquisition of

[*]Contributed during internship in Microsoft Research
[2]Microsoft Research, Cambridge, UK
[†]Now at Google AI, Berlin, Germany (contributed while being with Microsoft Research)
Correspondence to: Cheng Zhang <Cheng.Zhang@microsoft.com> and Wenbo Gong <wg242@cam.ac.uk>

training data. Intuitively, unfamiliar, but informative features are more useful for model training. We refer to this as element-wise training-time active acquisition.

Training-time active acquisition is needed in a great range of applications. One example is the recommender system with no historical user data.

Despite the success of element-wise test-time active prediction [23, 28, 44, 56], few works have provided a general and scalable solution for the *ice-start* problem. Additionally, these works [21, 22, 32] are commonly limited to a specific application scenario. More importantly, we need to design new acquisition functions that take the model parameters uncertainty into account.

In this work, we propose *Icebreaker* [1], a principled and efficient framework to solve the ice-start problem. *Icebreaker* actively acquires informative feature elements during training and also performs two general test tasks. To enable Icebreaker, we contribute the following:

1. We propose a Bayesian deep Latent Gaussian Model (BELGAM). Standard training of the deep generative model produces the point estimates for the parameters, whereas our approach applies a fully Bayesian treatment to the weights. The resulting epistemic uncertainty can be later used for training acquisition. (Section 2)

2. We design a novel partial amortized inference method for BELGAM, named PA-BELGAM. We combine the efficient amortized inference for the local latent variables with stochastic gradient MCMC for the model parameters to ensure high inference accuracy. (Section 2.2)

3. To complete Icebreaker, we propose two training-time element-wise information acquisition functions based on PA-BELGAM for imputation (Section 3) and active prediction (Section 4) tasks, respectively.

4. We evaluate PA-BELGAM and the entire Icebreaker approach on common machine learning benchmarks and a real-world health-care task. Our method demonstrates clear improvements when compared to multiple baselines, showing that it can be effectively used to solve the ice-start problem. (Section 5)

## 2 Bayesian Deep Latent Gaussian Model (BELGAM) with Partial Amortized Inference

Here, we propose a Bayesian Deep Latent Gaussian Model (BELGAM) with explicit epistemic uncertainty quantification, and a novel hybrid inference scheme for efficient and accurate inference.

### 2.1 Bayesian Deep Latent Gaussian Model (BELGAM)

A Bayesian latent variable model shown in Figure 1, is a common modeling choice, but previous work has focused on models that are typically linear and not flexible enough to model highly complex data. On the other hand, Deep Latent Gaussian Model [20], which uses a flexible neural network, does not quantify the parameter uncertainty. We unify the above two models and propose a Bayesian Deep Latent Gaussian Model (BELGAM), which uses a Bayesian neural network to generate observations $X_O$ from local latent variables $Z$ with global weights $\theta$ shown in Figure 1. The model is thus defined as:

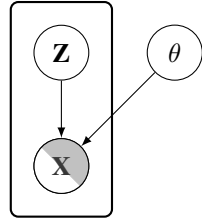

Figure 1: BELGAM

$$p(\boldsymbol{X}_O, \theta, \boldsymbol{Z}) = p(\theta) \prod_{i=1}^{|O|} \prod_{d \in O_i} p(x_{i,d}|\boldsymbol{z}_i, \theta) p(\boldsymbol{z}_i), \qquad (1)$$

where $|O|$ is the amount of observed data, and $O_i$ is the et of indices of observed feature entries for the $i$th data point. The goal is to infer the posterior, $p(\theta, \boldsymbol{Z}|\boldsymbol{X}_O)$, for both local latent variables $\boldsymbol{Z} = [\boldsymbol{z}_1, \ldots, \boldsymbol{z}_{|o|}]$ and global latent weights $\theta$. However, the posterior is generally intractable, and approximate inference is needed [25, 57]. Variational inference (VI) [3, 18, 25, 52, 57] and sampling-based methods [1] are two types of approaches commonly used for this task. Sampling-based approaches are known for accurate inference performances and theoretical guarantees[6].

However, sampling the local latent variable $\boldsymbol{Z}$ is computationally expensive as the cost scales linearly with the data set size. To best trade off computational cost against inference accuracy, we propose to amortize the inference for $\boldsymbol{Z}$ and keep an accurate sampling-based approach for the global latent weights $\theta$. Specifically, we use preconditioned stochastic gradient Hamiltonian Monte Carlo (SGHMC) [6] (see appendix for details).

## 2.2 Partial Amortized BELGAM

**Revisiting amortized inference in the presence of missing data.** Amortized inference [20, 38] is an efficient extension for variational inference. It was originally proposed for inferring local latent variables $\boldsymbol{Z}$ of deep latent Gaussian models. Amortized inference uses a deep neural network as a function estimator to compute the variational distribution $q(\boldsymbol{z}_i|x_i)$ for the posterior of $\boldsymbol{z}_i$ using $\boldsymbol{x}_i$ as input, instead of using individually parameterized approximations $q(\boldsymbol{z}_i)$. Thus, the estimation of the local latent variable does not scale with data set size during model training.

However, in our problem setting, the feature values for each data instance are partially observed. Thus, the vanilla amortized inference approach cannot be used as the input dimensionality of the observed data can vary for each data instance. As with the Partial VAE proposed in [28], we adopt a set encoding structure [37, 55] to build an inference network to infer $\boldsymbol{Z}$ based on partial observations in an amortized manner.

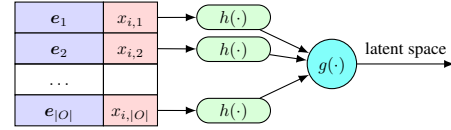

Figure 2: The illustration of P-VAE inference network structure.

The structure of the inference net is shown in Figure 2. For each data instance $\boldsymbol{x}_i \in \boldsymbol{X}_O$ with $|O_i|$ observed features, the input is modified as $\boldsymbol{S}_i = [\boldsymbol{s}_{i,1}, \ldots, \boldsymbol{s}_{i,|O_i|}]$ where $\boldsymbol{s}_{i,d} = [x_{i,d}, \boldsymbol{e}_d]$ and $\boldsymbol{e}_d$ is a feature embedding. This is fed into a standard neural network $h \colon \mathbb{R}^{M+1} \to \mathbb{R}^K$ where $M$ and $K$ are the dimensions of the latent space and $\boldsymbol{e}_d$, respectively. Finally, a permutation invariant set function $g(\cdot)$ is applied.

**Amortized inference + SGHMC** As discussed previously, we want to be computationally efficient when inferring $\boldsymbol{Z}$ and be accurate when inferring the global latent weights $\theta$ for BELGAM. Here, we discuss how to combine an accurate sampling approach for the global parameters with the efficient amortized inference for the local latent variables.

Assume we have the factorized approximated posterior $q(\theta, \boldsymbol{Z}|\boldsymbol{X}_O) \approx q(\theta|\boldsymbol{X}_O)q_\phi(\boldsymbol{Z}|\boldsymbol{X}_O)$ [20, 28], then the proposed inference scheme can be summarized into two stages: (i) Sample $\theta \sim q(\theta|\boldsymbol{X}_O)$ using SGHMC, (ii) Update the amortized inference network $q_\phi(\boldsymbol{z}_i|\boldsymbol{x}_i)$ to approximate $p(\boldsymbol{z}_i|\boldsymbol{x}_i)$.

First, we present how to sample $\theta \sim q(\theta|\boldsymbol{X}_O)$ using SGHMC. The optimal form for $q(\theta|\boldsymbol{X}_O)$ can be defined as $q(\theta|\boldsymbol{X}_O) = \frac{1}{C} e^{\log p(\boldsymbol{X}_O, \theta)}$, where $C$ is the normalization constant $p(\boldsymbol{X}_O)$. The key to sampling from such distribution is to compute the gradient $\nabla_\theta \log p(\boldsymbol{X}_O, \theta)$, which, unfortunately, is intractable due to marginalizing the latent variable $\boldsymbol{Z}$. Instead, we propose to approximate this quantity by transforming the marginalization into an optimization:

$$\log p(\boldsymbol{X}_O, \theta) \geq \sum_{i \in \boldsymbol{X}_O} \left[ \mathbb{E}_{q_\phi(\boldsymbol{z}_i|\boldsymbol{x}_i)}[\log p(\boldsymbol{x}_i|\boldsymbol{z}_i, \theta)] - KL[q_\phi(\boldsymbol{z}_i|\boldsymbol{x}_i)||p(\boldsymbol{z}_i)] \right] + \log p(\theta), \quad (2)$$

where right hand side is the lower bound of the joint distribution. Assuming that $\mathcal{F}$ is a sufficiently large function class, we can compute the gradient as:

$$\nabla_\theta \log p(\boldsymbol{X}_O, \theta) = \nabla_\theta \max_{q_\phi \in \mathcal{F}} \sum_{i \in \boldsymbol{X}_O} \left[ \mathbb{E}_{q_\phi(\boldsymbol{z}_i|\boldsymbol{x}_i)}[\log p(\boldsymbol{x}_i|\boldsymbol{z}_i, \theta)] - KL[q_\phi(\boldsymbol{z}_i|\boldsymbol{x}_i)||p(\boldsymbol{z}_i)] \right] + \log p(\theta).$$

$$(3)$$

After sampling $\theta$, we then update the inference network with these samples by optimizing:

$$\mathcal{L}(\boldsymbol{X}_O; \phi) = \mathbb{E}_{q(\theta, \boldsymbol{Z}|\boldsymbol{X}_O)}[\log p(\boldsymbol{X}_O|\boldsymbol{Z}, \theta)] - KL[q(\boldsymbol{Z}, \theta|\boldsymbol{X}_O)||p(\boldsymbol{Z}, \theta)]$$

$$= \mathbb{E}_{q(\theta|\boldsymbol{X}_O)} \left[ \sum_{i \in \boldsymbol{X}_O} \mathbb{E}_{q_\phi(\boldsymbol{z}_i|\boldsymbol{x}_i)}[\log p(\boldsymbol{x}_i|\boldsymbol{z}_i, \theta)] - KL[q_\phi(\boldsymbol{z}_i|\boldsymbol{x}_i)||p(\boldsymbol{z}_i)] \right] - KL[q(\theta|\boldsymbol{X}_O)||p(\theta)].$$

$$(4)$$

where the outer expectation can be approximated by SGHMC samples, and the outer KL penalty is intractable but can be ignored for updating the inference network. The resulting inference algorithm

resembles an iterative update procedure, like *Monte Carlo Expectation Maximization* (MCEM) [53] where it samples latent $\mathbf{Z}$ and optimizes $\theta$ instead. We call the proposed model *Partial Amortized BELGAM* (*PA-BELGAM*). Partial VAE [27] is actually a special case of PA-BELGAM, where $\theta$ is estimated by a point instead of with a set of samples.

Note that, in this way, the computational cost with the single-chain SGHMC is exactly the same as training a normal VAE thanks to the amortization for $\mathbf{Z}$. Thus, PA-BELGAM scales to large data when needed. For additional memory cost, we adopt a similar idea based on the Moving Window MCEM algorithm [12], where samples are stored and updated in a fixed size pool with a first in first out procedure. In the next two sections, we present two objective functions for two general machine learning tasks respectively: imputation tasks and prediction tasks.

## 3   Icebreaker for Imputation Tasks
We present Icebreaker for imputation tasks, which can be directly applied in the same way as [27].

**Problem Definition**   Assume that at each training data acquisition step we have already obtained training data $\mathcal{D}_{train}$, a pool data set $\mathcal{D}_{pool}$ that contains the data we could query next and $\mathcal{D}_{train} \cup \mathcal{D}_{pool} = \mathbf{X} \in \mathbb{R}^{N \times D}$. In the ice-start scenario, $\mathcal{D}_{train} = \emptyset$. At each step of the training-time acquisition, we actively select data points $x_{i,d} \in \mathcal{D}_{pool}$ to acquire, thereby moving them into $\mathcal{D}_{train}$ and updating the model with the newly formed $\mathcal{D}_{train}$. Figure 3 shows the flow diagram of this procedure at a given step. During the process, there is an observed data set $\mathbf{X}_O$ (e.g. the training data set $\mathbf{X}_O = \mathcal{D}_{train}$) and unobserved set $\mathbf{X}_U$ with $|O|$ and $|U|$ number of rows respectively. For each data instance $\boldsymbol{x}_i \in \mathbf{X}_O$, we have the observed index set $O_i$ containing the indices of the observed features for row $i$. The training time acquisition procedure is summarised in algorithm 1.

---

**Algorithm 1:** Element-wise training time acquisition

---
**input :** $\mathbf{X}_O, \mathbf{X}_U, \Phi, \mathcal{M}$, Acquisition number $K$, $\Xi$
$\mathbf{X}_O = \emptyset$;
**while** $\mathbf{X}_U \neq \emptyset$ **do**
    /* Information acquisition */
    Compute reward $R(x_{i,d}, \mathbf{X}_O)$ for $x_{i,d} \in \mathbf{X}_U$ using
     Eq. 5 or 10 ;    // Reward computation
    Sample $\mathbf{X}_{new}$ ; // Sample $K$ feature elements
     according to the $R$ value.
    $\mathbf{X}_O = \mathbf{X}_O \cup \mathbf{X}_{new}$;    // Update training set
    /* Model Training */
    Re-initialize model $\mathcal{M}$ ;    // Re-initialization
     to avoid local optimum
    $\mathcal{M} =$Train($\mathcal{M}, \Xi$);
    /* Test task */
    Test($\mathcal{M}$);     // Test performance of the
     current model $\mathcal{M}$
**end**

---

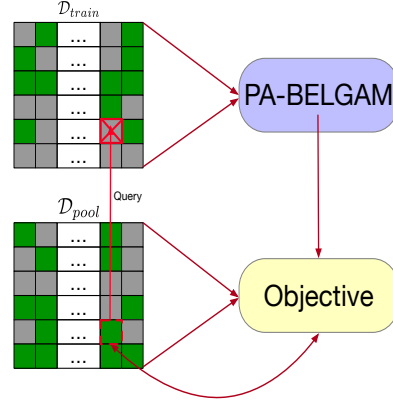

Figure 3: Icebreaker Flowchart. The green and gray blocks represent observed and unobserved items respectively.

We denote the training set $\mathcal{D}_{train} = \mathbf{X}_O$ and the pool set $\mathcal{D}_{pool} = \mathbf{X}_U$. The model $\mathcal{M}$ and training hyper-parameters are grouped as $\Xi$. We evaluate its quality on the test task using metrics such as predictive negative log likelihood (NLL).

### 3.1   Active Information Acquisition for Imputation

Designing the training time acquisition function is nontrivial. Existing information-theoretical objectives such as the one used in EDDI is not applicable in this setting (see appendix C.1). The key for such an objective function is to make the model certain about the data set as quickly as possible simultaneously focus on improving test performance.

Imputing missing values is important in applications such as recommender systems and other downstream tasks. In this setting, the goal is to learn about all the feature elements as quickly as possible. This can be formalized as selecting the elements $x_{i,d}$ that maximize the expected reduction in the posterior uncertainty of $\theta$:

$$R_I(x_{i,d}, \mathbf{X}_O) = H[p(\theta|\mathbf{X}_O)] - \mathbb{E}_{p(x_{i,d}|\mathbf{X}_O)}[H[p(\theta|\mathbf{X}_O, x_{i,d})]], \tag{5}$$

where $H[\cdot]$ denotes the entropy of a distribution. We use the symmetry of the mutual information to sidestep the posterior update $p(\theta|\boldsymbol{X}_O, x_{i,d})$ and entropy estimation of $\theta$ for efficiency. Thus, Eq. 5 is written as

$$R_I(x_{i,d}, \boldsymbol{X}_O) = H[p(x_{i,d}|\boldsymbol{X}_O)] - \mathbb{E}_{p(\theta|\boldsymbol{X}_O)}[H[p(x_{i,d}|\theta, \boldsymbol{X}_O)]]. \quad (6)$$

We can approximate Eq. 6 as

$$R_I(x_{i,d}, \boldsymbol{X}_O) \approx -\frac{1}{K} \sum_k \log \frac{1}{MN} \sum_{m,n} p(x_{i,d}^k|\boldsymbol{z}_i^m, \theta^n) + \frac{1}{NK} \sum_{k,n} \log \frac{1}{M} \sum_m p(x_{i,d}^k|\boldsymbol{z}_i^m, \theta^n), \quad (7)$$

based on the samples $\{\theta^n\}_{n=1}^N$, $\{\boldsymbol{z}_i^m\}_{m=1}^M$ and $\{x_{i,d}^k\}_{k=1}^K$ from SGHMC, the amortized inference network and the data distribution, respectively. The sample $x_{i,d} \sim p(x_{i,d}|\boldsymbol{X}_O)$ can be generated in the following way: (i) $\boldsymbol{z}_i \sim q_\phi(\boldsymbol{z}_i|\boldsymbol{x}_{io})$, (ii) $\theta \sim q(\theta|\boldsymbol{X}_O)$ and (iii) $x_{i,d} \sim p(x_{i,d}|\theta, \boldsymbol{z}_i)$, where $\boldsymbol{x}_{io}$ represents the observed features in the $i^{th}$ row of $\boldsymbol{X}_O$

## 4 Icebreaker for Prediction Tasks

Next, we introduce a second type of test task called active prediction, where a sequence of active acquisition steps is carried out before predicting a specified target variable at test time. Note that the typical test prediction task is a special case where no acquisition of features is performed. Here, we demonstrate the case where feature-wise active information acquisition is used in both training and testing time, which is desired in data costly situations.

**Problem Definition**  During the training acquisition, the procedure is the same as in the imputation task, which is shown in Algorithm 1 and Figure 3. The only difference is that we have specified target variables. We denote the target as $\boldsymbol{Y}$. In this case, each $\boldsymbol{x}_i \in \boldsymbol{X}_O$ has a corresponding target $\boldsymbol{y}_i$. In addition, instead of querying a single feature value $x_{i,d}$ during training, as in the imputation task, we query a feature-target pair $(x_{i,d}, y_i)$ if $y_i$ has not been queried before. Otherwise, we only query $x_{i,d}$. As an example, we adopt a similar procedure used in EDDI [28] for test time active prediction, and use the *Area under the information curve* (AUIC) generated from EDDI to evaluate the performance of Icebreaker. This reflects the overall model performance with test time active acquisition. The evaluation procedure is summarised in Algorithm 3 in the appendix.

### 4.1 Model and Active Information Acquisition for Active Prediction

**Conditional BELGAM**  The proposed model and inference algorithm in section 3 can be easily extended to incorporate the target variables. In general, PA-BELGAM can be directly adapted to any VAE based framework. One possible choice is to adopt the formulation of the conditional VAE [45] for the prediction task here (see appendix B for details).

**Icebreaker for active target prediction.**  For the prediction task, solely reducing the model epistemic uncertainty is not optimal as the goal is to predict the target variable $\boldsymbol{Y}$. Instead, we require the model to (1) capture feature correlations for accurate imputations in both training and test time (similar to reducing the model epistemic uncertainty), and (2) find informative features to learn to predict the target variable. Thus, the desired acquisition function needs to balance the unsupervised learning, which focuses on exploring relations between features, and supervised learning that exploits informative features to predict specified targets. We propose the following objective:

$$R_P(x_{i,d}, \boldsymbol{X}_O) = \mathbb{E}_{p(x_{i,d}|\boldsymbol{X}_O)}[H[p(\boldsymbol{y}_i|x_{i,d}, \boldsymbol{X}_O)]] - \mathbb{E}_{p(\theta, x_{i,d}|\boldsymbol{X}_O)}[H[p(\boldsymbol{y}_i|\theta, x_{i,d}, \boldsymbol{X}_O)]]. \quad (8)$$

The above objective is the *conditional mutual information* $I(\boldsymbol{y}_i, \theta|x_{i,d}; \boldsymbol{X}_O)$. Thus, maximizing 8 is the same as maximizing the information gain between the target $y_i$ and the model weights $\theta$, conditioned on the additional feature $x_{i,d}$, and observed features $X_O$. In our case, the $x_{i,d}$ is unobserved. As the weights $\theta$ do not change significantly after collecting $x_{i,d}$, for computational convenience, we assume $p(\theta|\boldsymbol{X}_O) \approx p(\theta|\boldsymbol{X}_O, x_{i,d})$ when estimating the objective.

As before, we approximate this objective using Monte Carlo integration:

$$R_P(x_{i,d}, \boldsymbol{X}_O) \approx$$
$$-\frac{1}{JK} \sum_{j,k} \log \frac{1}{MN} \sum_{m,n} p(\boldsymbol{y}_i^{(j,k)}|\boldsymbol{z}_i^{(m,k)}, \theta^n) + \frac{1}{KNJ} \sum_{j,n,k} \log \frac{1}{M} \sum_m p(\boldsymbol{y}_i^{(j,k)}|\boldsymbol{z}_i^{(m,k)}, \theta^n), \quad (9)$$

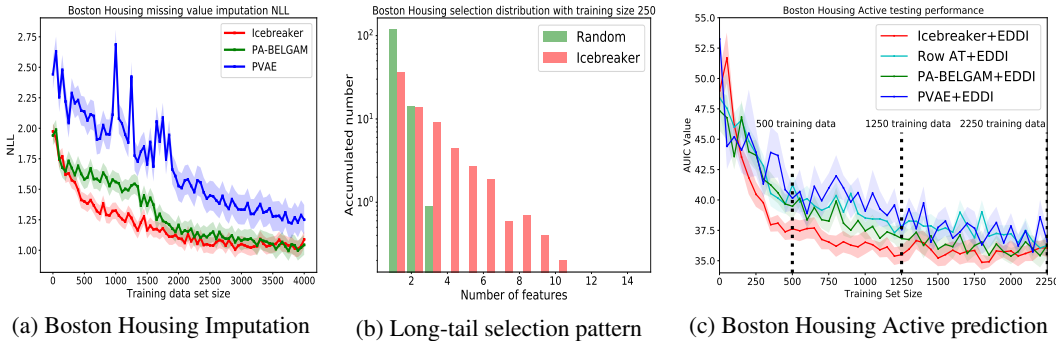

| (a) Boston Housing Imputation | (b) Long-tail selection pattern | (c) Boston Housing Active prediction |

Figure 4: Boston Housing experimental results. (a) The NLL over the number of observed feature values. (b) The distribution (log scale) of the number of observed features per data instance during the training time. (c) Performance on the active prediction task vs. training set size. The test time active prediction curves at the training data size indicated by the black dash line are shown in Figure 5

where we draw $\{\boldsymbol{z}_i^{(m,k)}\}_{m=1}^M$ from $q_\phi(\boldsymbol{z}_i|\boldsymbol{X}_O, x_{i,d}^k)$ for each imputed sample $x_{i,d}^k$. Others ($\{\theta^n\}_{n=1}^N$, $\{\boldsymbol{y}_i^{(j,k)}\}_{j=1}^J$ and $\{x_{i,d}^k\}_{k=1}^K$) are sampled in a similar way as in the imputation task. This objective naturally balances the exploration of new unseen features that may be informative as well as the exploitation of the familiar ones to facilitate learning a better predictor. For example, if feature $x_{i,d}$ has not been observed before or uninformative about the target, the first entropy term in Eq. 8 will be high, which encourages the algorithm to pick this data point. However, using this term alone may result in selecting uninformative/noisy features. Thus, we need an extra term that eliminates the possibility of selecting uninformative features, which is exactly the second term. Unless $x_{i,d}$ together with $\theta$ can provide extra information about $\boldsymbol{y}_i$, the entropy in the second term for uninformative features will still be high. Thus, the two terms combined together encourage the model to select the less explored but informative features. The resulting objective is mainly targeted at (2) mentioned at the beginning of this subsection. Thus, a natural way to satisfy both (1) and (2) is a combination of the two objectives:

$$R_C(x_{i,d}, \boldsymbol{X}_O) = (1-\alpha)R_I(x_{i,d}, \boldsymbol{X}_O) + \alpha R_P(x_{i,d}, \boldsymbol{X}_O), \tag{10}$$

where $\alpha$ controls which task the model focuses on. This objective also has an information-theoretic interpretation. In the appendix C.1, we show that when $\alpha = \frac{1}{2}$, this combined objective is equivalent to the mutual information between $\theta$ and the feature-target pair $(x_{i,d}, \boldsymbol{y}_i)$.

## 5 Experiments

We evaluate Icebreaker first on benchmark data sets UCI [8] on both imputation and prediction tasks. We then consider two real-world applications: (a) movie rating imputation task using the *MovieLens* dataset [10]; and (b) risk prediction in intensive care using the MIMIC dataset [17].

**Experiments Setup and evaluation.** We compare Icebreaker with a random feature acquisition strategy for training where both P-VAE [28] and PA-BELGAM are used. For the imputation task, P-VAE already achieves excellent results in various data sets compared to traditional methods [28, 34]. Additionally, for the active prediction task, we compare Icebreaker to an instance-wise active learning method, denoted as *Row AT*, in which the data are assumed to be fully observed apart from the target.

We evaluate the imputation performance by reporting *negative log likelihood* (NLL) over the test target. For the active prediction task, we use EDDI [28] to sequentially select features at test time. We report the *area under the information curve* (AUIC) [28] for the test set (See Figure 5 for an example and the appendix for details). A smaller value of AUIC indicates better overall active prediction performance. All experiments are averaged over 10 runs, and their setting details are in the appendix.

### 5.1 UCI Data Set

**Imputation Task.** At each step of Icebreaker, we select 50 feature elements from the pool. Figure 4a shows the averaged NLL on the test set as the training set increases. Icebreaker outperforms

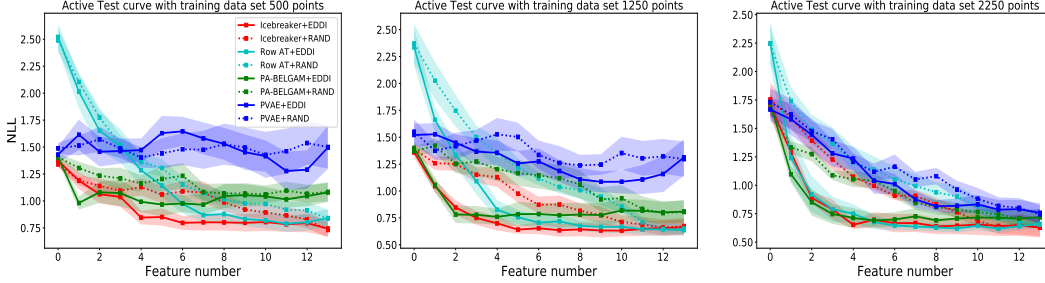

Figure 5: Evaluation of test time performance after exposure to different amounts of training data: (**Left**): 550 feature elements. (**Middle**):1250 feature elements (**Right**): 2250 feature elements. The x-axis indicates the number of actively-acquired feature elements used for prediction. Legend indicates the methods used for training (Icebreaker, Row AT, etc.) and test time acquisition (EDDI, RAND)

random acquisition with both PA-BELGAM and P-VAE by a large margin, especially at the early stages of training. We also see that PA-BELGAM alone can be beneficial compared to P-VAE with small data sets. This is because P-VAE tends to over-fit, while PA-BELGAM leverages the model uncertainties.

We also analyze the selection pattern. We gather all the rows that have been queried with at least one feature during training acquisition and count how many features are queried for each. We repeat this for the first 5 acquisitions. Figure 4b shows the histogram of the number of features acquired for each data point. The random selection concentrates around one feature per data instance. However, the long-tailed distribution of the number of features selected by Icebreaker means it tends to concentrate more features in certain rows to exploit feature relations for predicting target but simultaneously tries to spread its selection for more exploration. We include imputation results on other UCI data sets in the Appendix. We find that Icebreaker consistently outperforms the baselines by a large margin.

**Prediction Task.** Figure 4c shows the AUIC curve as the amount of training data increases. The Icebreaker clearly achieves better results compared to all baselines (Also confirmed by Figure 5). This shows that it not only yields a more accurate prediction of the targets but also captures correlations between features and targets. Interestingly, the baseline *Row AT* performs a little worse than PA-BELGAM. We argue that before querying a single target variable, *Row AT* needs to query the whole row, which induces the costs equivalent to the number of features. Thus, with fixed query budgets, *Row AT* will form a relatively small but complete data set. Again, the uncertainty of PA-BELGAM brings benefits compared to P-VAE with point estimated parameters.

At the early training stage (500 data points, the left panel in Figure 5), the performance of *Row AT* is worse at test time than others when few features are selected. This is due to the fact that obtaining a complete observed datum is costly. With the budget of $500$ feature elements, it can only select $50$ fully observed data instances. In contrast, Icebreaker has obtained, within that budget, 260 partially observed instances with different levels of missingness. As more features are selected during the test, these issues are mitigated, and the performance starts to improve. Further evidence suggests that, as the training data grows, we can clearly observe a better prediction performance of *Row AT* at the early test stage. We also include in the appendix the evaluation of other UCI data sets for active prediction.

## 5.2 Recommender System using MovieLens

One common benchmark data set for recommender systems is *MovieLens-1M* [10]. P-VAE has obtained state-of-the-art imputation performance in this dataset after training with a sufficient amount of data [27]. Figure 6a shows the performance on predicting unseen data points in terms of NLL. Icebreaker shows that with minimum training data, the model has already learned to predict the unseen data with high accuracy. Given any small amount of data, Icebreaker obtains the best performance at the given query budget, followed by PA-BELGAM which outperforms P-VAE. The selection pattern in Figure 6b is similar to the UCI imputation, shown in Figure 6b. We argue this long-tail selection is important, especially when each row contains many features. The random selection tends to scatter the choices and is less likely to discover dependencies until the data set grows larger. However, if

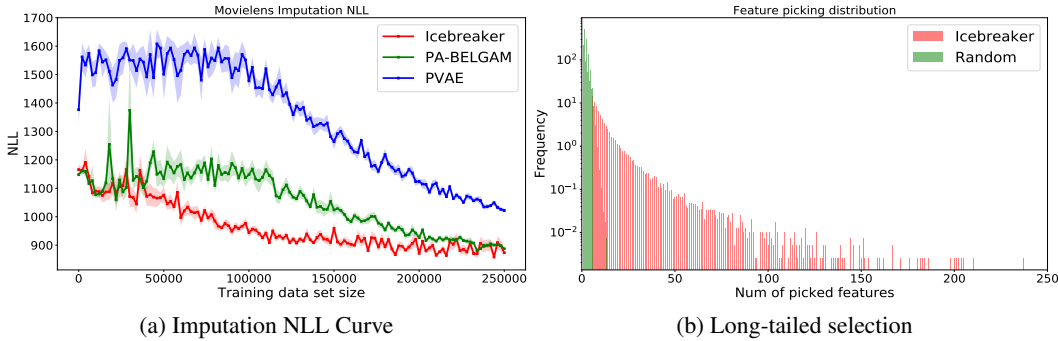

(a) Imputation NLL Curve                    (b) Long-tailed selection

Figure 6: Performance on MovieLens. Panel (a) shows the imputation NLL vs. the number of observed movie ratings. Panel (b) shows the distribution of the number of features selected per user.

there are many features per data instance, this accumulation will take a very long time. On the other hand, the long-tailed selection exploits the features inside certain rows to discover their dependencies and simultaneously tries to spread out the queries for exploration.

## 5.3  Mortality Prediction using MIMIC

We apply Icebreaker in a health-care setting using the Medical Information Mart for Intensive Care (MIMIC III) data set [17]. This is the largest real-world health-care data set in terms of patient numbers. The goal is to predict mortality based on 17 medical measurements. The data is pre-processed following [11] and balanced. Full details are available in appendix E.2.1.

The left panel in Figure 7 shows that the Icebreaker outperforms the other baselines significantly in active prediction with higher robustness (smaller std. error). Robustness is crucial in health-care settings as the cost of unstable model performance is high. As before, *Row AT* performs worse until it accumulates sufficient data. Note that without active training feature selection, PA-BELGAM performs better than P-VAE due to its ability to model uncertainty, which is very useful in this extremely noisy data set.

To evaluate whether the proposed method can discover valuable information, we plot the accumulated feature number in the middle panel of Figure 7. The x-axis indicates the total number of observed data in the training set, and each point on the curve indicates the number of features selected in the corresponding training set. We see that not only different features have been collected at different frequencies, but the curve of Glucose is clearly non-linear as well. This indicates that the importance of different features varies for different training set size. Icebreaker is establishing a sophisticated feature element acquisition scheme that no heuristic method can currently achieve.   The top 3 features are the *Glasgow coma scale* (GCS). These features have been identified previously as being clinically important (e.g. by the IMPACT model [47]). *Glucose* is also in the IMPACT set. It was not collected frequently in the early stage, but in the later training phase, more *Glucose* feature has been selected.  Compared to GCS, Glucose has a highly non-linear relationship with the patient outcome [36] (or refer to the appendix E.2.1). Icebreaker chooses more informative features with simpler relationships in the very early iterations. While the learning progresses, Icebreaker is able to identify these informative features with complex relationships to the target. Additionally, the missing rate for each feature in the entire data set differs. *Capillary refill rate* (*Cap.*) has more than $90\%$ data missing, much higher than *Height*. Icebreaker is still able to pick the useful and rarely observed information, while only choosing a small percent of the irrelevant information at test time.  On the right hand side of Figure 7, we plot the histogram of the initial choices during test-time acquisition. *GCS* are mostly selected in the first step, as it is the most informative feature.

## 6  Related Work

**Data-wise Active Learning.**   The goal of active learning is to obtain optimal model performance with as fewer queries as possible [29, 31, 43], where only querying labels are associated with a cost. One category is based on decision theory [39], where the acquisition step is to minimize the

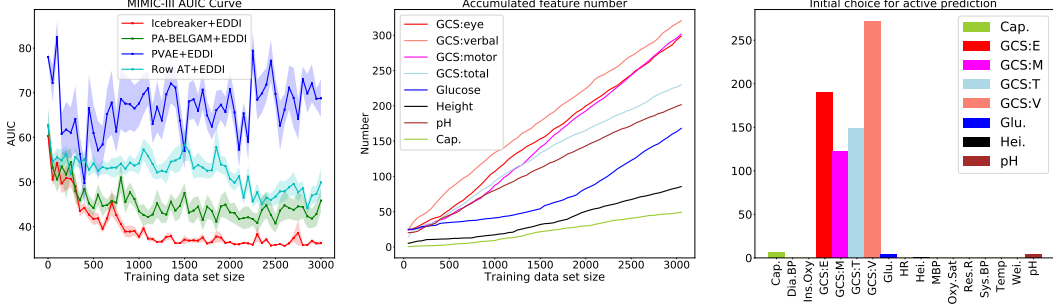

Figure 7: Performance MIMIC experiments. (**Left**) This figure shows the predictive AUIC curve as training data size increases. (**Middle**) The accumulated feature statistics as active selection progresses (**Right**) This indicates the histogram of initial choice during active prediction using EDDI.

loss defined by test tasks after making the query based on observed data. Indeed this coincides perfectly with the goal of active learning. However, its evaluation can be expensive in practice [19, 59]. Another category is based on information theory, including many previous active learning approaches [7, 26, 50]. Another well-known acquisition function is BALD [14], which is based on mutual information. Although our acquisition for imputation is also based on mutual information, we emphasize that the original BALD objective is only applied to scenarios with complete data set. In another word, those methods aim to only select next data instance to label while assuming that every feature of each data point is observed. We call this approach *instance-wise* selection. Obviously, these methods are not directly applicable to the *ice-start* problem as they assume that the only cost comes from acquiring labels.

**Feature-wise Active Learning.** Instead of only querying labels, the above active learning idea can be extended to query features, named as active feature acquisition (AFA). It makes sequential feature selections in order to improve model performance[5, 15, 32, 40, 41, 48, 49], which is similar to our framework. However, they are commonly designed for a specific application such as clustering [51] and classification [33], assuming the data are fully observed in the test time. In addition, many methods have other limitations. For example, only simple linear models can be used [5, 40, 48] with non-information-theoretical objective functions [15, 32]. None of the above methods can be easily combined with test time active prediction methods [16, 28, 44]. Our method enables both training time and test-time efficient information acquisition in a principled way with a flexible model, which is of great need in real-life applications.

**Cold-start problem** Another relevant problem to *ice-start* is called *cold-start* problem [30, 42]. The key difference between these two scenarios is that cold-start problem targets at the test time data scarcity after the model has been trained. Taking the recommender system as an example, the cold-start problem handles the scenario when there are new users incoming with no historical ratings given a trained recommender. One common strategy is to utilise the meta data (e.g. user profiles, item category) to initialise the latent factors of users/items [35, 46, 54].

## 7    Conclusion

In this work, we introduce the ice-start problem where machine learning models are expected to be deployed where little or no training data has been collected. The costs of collecting new training datum apply at the level of feature elements. Icebreaker provides an information-theoretical way to acquire element-wise data for training actively and uses the minimum amount of data for downstream test tasks like *imputation* and *active prediction*. Within the framework of *Icebreaker*, we propose PA-BELGAM, a Bayesian deep latent Gaussian model together with a novel inference scheme that combines amortized inference and SGHMC. This enables fast and accurate posterior inference. Furthermore, we propose two training time acquisition functions targeted at the *imputation* and *active prediction* tasks. We evaluate *Icebreaker* on several benchmark data sets, including two real-world applications. Icebreaker consistently outperforms the baselines. Possible future directions include taking the mixed-type variables into account and deploying it in a pure streaming environment.

## Footnotes

[1]Department of Engineering, University of Cambridge, Cambridge, UK

[1]Code available: `https://github.com/microsoft/Icebreaker`

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
