[Supplementary Material · Icebreaker_CameraReady_supp.pdf]

# A  Stochastic Gradient HMC

Assume we want to draw samples $\theta \sim p(\theta|\boldsymbol{X}_O)$, the potential energy $U(\theta)$ is defined as (for our partial amortized inference algorithm, this is replaced with Eq. 3)

$$U(\theta) = - \sum_{\boldsymbol{x}_I \in \boldsymbol{X}_O} \log p(\boldsymbol{x}_i|\theta) - \log p(\theta). \qquad (11)$$

Typically, we use the mini-batch to estimate this quantity, therefore, the stochastic estimate of $U(\theta)$ with the batch $\boldsymbol{X}_S$ can be written as

$$\tilde{U}(\theta, \boldsymbol{X}_S) = -\frac{|O|}{|S|} \sum_{\boldsymbol{x}_i \in \boldsymbol{X}_S} \log p(\boldsymbol{x}_i|\theta) - \log p(\theta), \qquad (12)$$

where $|S|$ and $|O|$ are the number of rows for $\boldsymbol{X}_S$ and $\boldsymbol{X}_O$ respectively.

The preconditioned SGHMC [6] uses the diagonal Fisher information matrix as the adaptive preconditioner with moving average approximations[24]. Thus, the transition dynamics at time $t$ is the following :

$$
\left.
\begin{array}{l}
B = \frac{1}{2}\epsilon \\
V_{t-1} = (1 - \tau)V_{t-2} + \tau \nabla_\theta \tilde{U}(\theta) \cdot \nabla_\theta \tilde{U}(\theta) \\
g_{t-1} = \frac{1}{\sqrt{\lambda + \sqrt{V_{t-1}}}}
\end{array}
\right\} \text{Preconditioning Computation}
$$

$$
\left.
\begin{array}{l}
\boldsymbol{p}_t = (1 - \epsilon\beta)\boldsymbol{p}_{t-1} - \epsilon g_{t-1}\nabla_\theta \tilde{U}(\theta) + \epsilon\frac{\partial g_{t-1}}{\partial \theta_{t-1}} + \sqrt{2\epsilon(\beta - B)}\eta \\
\theta_t = \theta_{t-1} + \epsilon g_{t-1}\boldsymbol{p}_{t-1}
\end{array}
\right\} \text{SGHMC Updates}
\qquad (13)
$$

where $\eta \sim \mathcal{N}(\boldsymbol{0}, \boldsymbol{I})$ and $\epsilon$ is the step size. [6] shows that the continuous-time dynamics of the above transitions can indeed preserve the stationary distribution $\pi(\theta) \propto \exp(-U(\theta))$. In practice, the update equation of the preconditioning SGHMC Eq. 13 is closely related to Adam optimizer as discussed in [6]. Intuitively, this can be regarded as a specially designed Adam with properly scaled Gaussian noise. Algorithm 2 shows the procedure of the partial amortized inference.

---

**Algorithm 2:** Amortized Inference + SGHMC

**input :** Data $\boldsymbol{X}_O$, step size $\epsilon$, friction $\beta$, thinning $\tau$, learning rate $\gamma$, initialized $\theta$, max sample size $N$
**Result:** Variational parameter $\phi$ and $\{\theta_n\}_{n=1}^N$
Model and sampler initialization;
counter=0;
**while** *not converged* **do**
    Sample minibatch $\boldsymbol{X}_S \in \boldsymbol{X}_O$;
    Random masking with mask $\boldsymbol{m}$: $\tilde{\boldsymbol{X}}_S = \boldsymbol{X}_S \times \boldsymbol{m}$;
    `/* Inference Network Update */`
    Compute $\mathcal{L}(\tilde{\boldsymbol{X}}_S; \phi)$ using Eq.4;
    $q_\phi$: Optimize($\mathcal{L}(\tilde{\boldsymbol{X}}_S; \phi)$;Adam;$\gamma$);
    `/* SGHMC step */`
    Compute $\tilde{U}(\theta)$ using Eq.3 with proper scale;
    $\theta$: Simulate dynamics Eq.13;
    `/* Update the sample pool */`
    **if** *counter$= K\tau$, where $K$ is any positive integer* **then**
        | $\{\theta_n\}$ =Update($\{\theta_n\}$,$\theta$,N);            `// Using FIFO procedure`
    **end**
    counter+=1;
**end**

---

# B  Conditional BELGAM

We follow the similar notations as in main text, but we have additional target sets $\boldsymbol{Y}_O$ and $\boldsymbol{Y}^*$ in observed training and test data respectively. By similar derivations in [45], we have

$$\log p(\boldsymbol{Y}_O|\boldsymbol{X}_O, \theta) \geq \sum_{i \in \boldsymbol{X}_O} \left[ \mathbb{E}_{q_\phi(\boldsymbol{z}_i|\boldsymbol{x}_i, y_i)}[\log p(y_i|\boldsymbol{z}_i, \theta)] - KL[q_\phi(\boldsymbol{z}_i|\boldsymbol{x}_i, y_i)||p(\boldsymbol{z}_i|\boldsymbol{x}_i)] \right] \qquad (14)$$

Note that the encoder proposed in *BELGAM* can handle variable-sized inputs, thus, we can make further approximation $p(\boldsymbol{z}_i|\boldsymbol{x}_i) \approx q_\phi(\boldsymbol{z}_i|\boldsymbol{x}_i)$. We call the right hand side of Eq. 14 as $\mathcal{L}_{conditional}(\boldsymbol{Y}_O; \phi)$. We should note that $\mathcal{L}_{conditional}(\boldsymbol{Y}_O; \phi)$ only focuses on prediction quality. On the contrary, successful active prediction, as discussed in main text, requires the model not only has a better target prediction but also capture the correlations between input features for sequential active decisions. Thus, in practice, we need to include the Eq. 2 as well. Thus during each SGHMC step, Eq. 2 is replaced with

$$\mathcal{L}(\{\boldsymbol{Y}_O, \boldsymbol{X}_O\}; \phi) = \beta \mathcal{L}_{conditional}(\boldsymbol{Y}_O; \phi) + (1 - \beta)\mathcal{L}_{joint}(\boldsymbol{X}_O; \phi) \tag{15}$$

where $\beta$ controls which tasks the model focuses on. When $\beta = 0.5$, we have

$$\log p(\{\boldsymbol{X}_O, \boldsymbol{Y}_O\}, \theta) \geq \mathcal{L}(\{\boldsymbol{X}_O, \boldsymbol{Y}_O\}; \theta) \tag{16}$$

with equality holds when $q_\phi(\boldsymbol{z}_i|\boldsymbol{x}_i) = p(\boldsymbol{z}_i|\boldsymbol{x}_i)$ and $q_\phi(\boldsymbol{z}_i|\boldsymbol{x}_i, y_i) = p(\boldsymbol{z}_i|\boldsymbol{x}_i, y_i)$. In experiment, we choose $\beta = 0.6$. We can also derive the equivalent form for Eq.4 using similar procedures for inference network update.

## C    Information acquisition

### C.1    Theoretical results

**Review: EDDI.**    For the active target prediction, model need to decide which feature should be queried for the purpose of predicting the target $\boldsymbol{Y}$ accurately in each test selection step. [28] proposes a reward function for this test task inspired by Bayesian experimental design [4, 26]. They propose to select the data point $x_{i,d}$ by maximizing:

$$R(x_{i,d}, \boldsymbol{X}_O) = \mathbb{E}_{x_{i,d} \sim p(x_{i,d}|\boldsymbol{X}_O)} \left[KL[p(\boldsymbol{y}_i|x_{i,d}, \boldsymbol{X}_O)||p(\boldsymbol{y}_i|\boldsymbol{X}_O)]\right]. \tag{17}$$

We find that this can be written as the mutual information between the target $\boldsymbol{y}_i$ and the candidate $x_{i,d}$:

$$R(x_{i,d}, \boldsymbol{X}_O) = H[p(\boldsymbol{y}_i|\boldsymbol{X}_O)] - \mathbb{E}_{p(x_{i,d}|\boldsymbol{X}_O)}[H[p(\boldsymbol{y}_i|\boldsymbol{X}_O, x_{i,d})]]. \tag{18}$$

Thus, maximizing this quantity is equivalent to finding the most informative feature to the predictive target variable $\boldsymbol{y}_i$. However, this is not a suitable acquisition function in the training time as it is built on the assumption that the model is well trained and able to find the true informative features. Specifically, from Eq.18, it should be noted that $x_{i,d}$ is irrelevant to the first term. Thus, maximizing this objective is equivalent to minimizing the expected entropy after observing $x_{i,d}$, or conditional entropy $H(\boldsymbol{y}_i|x_{i,d})$. This objective purely encourages exploitation. For example, it can fail in the following scenario. In the beginning of training acquisition, the model may capture the wrong informative feature due to the small training data set. The exploitation nature of EDDI tends to pick this wrong feature over others in the following acquisitions and will be trapped into the sub-optimal strategy.

**EDDI for PA-BELGAM.**    Next, we show that with a trained *PA-BELGAM*, the above objective can be approximated efficiently. We assume the decoupled posterior $p(\theta, \boldsymbol{Z}|\boldsymbol{X}_o) \approx p(\theta|\boldsymbol{X}_O)p(\boldsymbol{Z}|\boldsymbol{X}_O)$ and conditionally independent features $p(\boldsymbol{x}_i|\boldsymbol{z}_i, \theta) = \prod_{d=1}^{|o_i|} p(x_{i,d}|\theta, \boldsymbol{z}_i)$. The EDDI rewards in Eq.17 can be rewritten by using KL chain rule:

$$KL[p(\boldsymbol{y}_i|x_{i,d}, \boldsymbol{X}_O)||p(\boldsymbol{y}_i|\boldsymbol{X}_O)] = KL[p(\boldsymbol{y}_i, \boldsymbol{z}_i, \theta|x_{i,d}, \boldsymbol{X}_O)||p(\boldsymbol{y}_i, \boldsymbol{z}_i, \theta|\boldsymbol{X}_O)]$$
$$- \mathbb{E}_{p(\boldsymbol{y}_i|x_{i,d}, \boldsymbol{X}_O)}[KL[p(\boldsymbol{z}_i, \theta|\boldsymbol{y}_i, x_{i,d}, \boldsymbol{X}_O)||p(\boldsymbol{z}_i, \theta|\boldsymbol{y}_i, \boldsymbol{X}_O)]]. \tag{19}$$

The first term can be further approximated as

$$KL[p(\boldsymbol{y}_i, \boldsymbol{z}_i, \theta|x_{i,d}, \boldsymbol{X}_O)||p(\boldsymbol{y}_i, \boldsymbol{z}_i, \theta|\boldsymbol{X}_O)] = KL[p(\boldsymbol{z}_i, \theta|x_{i,d}, \boldsymbol{X}_O)||p(\boldsymbol{z}_i, \theta|\boldsymbol{X}_O)]$$
$$+ KL[p(\boldsymbol{y}_i|\boldsymbol{z}_i, \theta, x_{i,d}, \boldsymbol{X}_O)||p(\boldsymbol{y}_i|\boldsymbol{z}_i, \theta, \boldsymbol{X}_O)]$$
$$= KL[p(\boldsymbol{z}_i|x_{i,d}, \boldsymbol{X}_O, \theta)||p(\boldsymbol{z}_i|\boldsymbol{X}_O, \theta)] + KL[p(\theta|x_{i,d}, \boldsymbol{X}_O)||p(\theta|\boldsymbol{X}_O)] + KL[p(\boldsymbol{y}_i|\boldsymbol{z}_i, \theta)||p(\boldsymbol{y}_i|\boldsymbol{z}_i, \theta)]$$
$$= KL[p(\boldsymbol{z}_i|x_{i,d}, \boldsymbol{X}_O)||p(\boldsymbol{z}_i|\boldsymbol{X}_O)]. \tag{20}$$

where the last equality holds if we assume no posterior updates for $\theta$. This is a reasonable assumption because $x_{i,d}$ is only single data point adding into a much larger set $\boldsymbol{X}_O$. By using similar trick, we can show the second term in Eq.19 is re-written as

$$
\begin{aligned}
&\mathbb{E}_{p(\boldsymbol{y}_i|x_{i,d},\boldsymbol{X}_O)}[KL[p(\boldsymbol{z}_i,\theta|\boldsymbol{y}_i,x_{i,d},\boldsymbol{X}_O)||p(\boldsymbol{z}_i,\theta|\boldsymbol{y}_i,\boldsymbol{X}_O)]] \\
&= \mathbb{E}_{p(\boldsymbol{y}_i|x_{i,d},\boldsymbol{X}_O)}[KL[p(\boldsymbol{z}_i|\boldsymbol{y}_i,x_{i,d},\boldsymbol{X}_O)||p(\boldsymbol{z}_i|\boldsymbol{X}_O,\boldsymbol{y}_i)] + KL[p(\theta|\boldsymbol{y}_i,\boldsymbol{X}_O,x_{i,d})||p(\theta|\boldsymbol{X}_O,\boldsymbol{y}_i)]] \\
&= \mathbb{E}_{p(\boldsymbol{y}_i|x_{i,d},\boldsymbol{X}_O)}[KL[p(\boldsymbol{z}_i|\boldsymbol{y}_i,x_{i,d},\boldsymbol{X}_O)||p(\boldsymbol{z}_i|\boldsymbol{X}_O,\boldsymbol{y}_i)]].
\end{aligned}
\tag{21}
$$

Then, we replace the posterior of $\boldsymbol{Z}$ with variational approximations $q_\phi$. Eq.17 can be aproximated as

$$
\begin{aligned}
R(x_{i,d},\boldsymbol{X}_O) \approx &\mathbb{E}_{p(x_{i,d}|\boldsymbol{X}_O)}[KL[q_\phi(\boldsymbol{z}_i|x_{i,d},\boldsymbol{X}_O)||q_\phi(\boldsymbol{z}_i|\boldsymbol{X}_O)]] \\
&- \mathbb{E}_{p(x_{i,d},\boldsymbol{y}_i|\boldsymbol{X}_O)}[KL[q_\phi(\boldsymbol{z}_i|\boldsymbol{y}_i,x_{i,d},\boldsymbol{X}_O)||q_\phi(\boldsymbol{z}_i|\boldsymbol{X}_O,\boldsymbol{y}_i)]].
\end{aligned}
\tag{22}
$$

This is exactly equivalent to the original form in [28]. The only difference is the sampling stage for $x_{i,d} \sim p(x_{i,d}|\boldsymbol{X}_O)$ and $x_{i,d}, \boldsymbol{y}_i \sim p(x_{i,d}, \boldsymbol{y}_i|\boldsymbol{X}_O)$, where the $\theta$ samples are needed.

$$
\begin{aligned}
\boldsymbol{z}_i &\sim q_\phi(\boldsymbol{z}_i|\boldsymbol{X}_O) \\
\theta &\sim p(\theta|\boldsymbol{X}_O) \quad \text{using SGHMC} \\
x_{i,d} &\sim p(x_{i,d}|\boldsymbol{z}_i,\theta) \\
\boldsymbol{y}_i &\sim p(\boldsymbol{y}_i|\boldsymbol{z}_i,\theta)
\end{aligned}
\tag{23}
$$

**Connections of Icebreaker acquisition function to mutual information** We now show that the information acquisition function proposed in Eq.10 with $\alpha = \frac{1}{2}$ is equivalent to the mutual information between $\theta$ and the feature-target pair $(\boldsymbol{y}_i, x_{i,d})$.

$$
\begin{aligned}
R_c(x_{i,d},\boldsymbol{X}_O) = &\underbrace{\frac{1}{2}H[p(x_{i,d}|\boldsymbol{X}_O)] + \frac{1}{2}\mathbb{E}_{p(x_{i,d}|\boldsymbol{X}_O)}[H[p(\boldsymbol{y}_i|x_{i,d},\boldsymbol{X}_O)]]}_{\textcircled{1}} \\
&\underbrace{-\frac{1}{2}\mathbb{E}_{p(\theta|\boldsymbol{X}_O)}[H[p(x_{i,d}|\theta,\boldsymbol{X}_O)]] - \frac{1}{2}\mathbb{E}_{p(\theta,x_{i,d}|\boldsymbol{X}_O)}[H[p(\boldsymbol{y}_i|\theta,x_{i,d},\boldsymbol{X}_O)]]}_{\textcircled{2}}.
\end{aligned}
\tag{24}
$$

For $\textcircled{1}$, we have

$$
\begin{aligned}
\textcircled{1} &= -\int p(x_{i,d}|\boldsymbol{X}_O)\left[\log p(x_{i,d}|\boldsymbol{X}_O) + \int p(\boldsymbol{y}_i|x_{i,d},\boldsymbol{X}_O)\log p(\boldsymbol{y}_i|x_{i,d},\boldsymbol{X}_O)d\boldsymbol{y}_i\right]dx_{i,d} \\
&= -\int p(x_{i,d}|\boldsymbol{X}_O)\int p(\boldsymbol{y}_i|x_{i,d},\boldsymbol{X}_O)\log p(x_{i,d},\boldsymbol{y}_i|\boldsymbol{X}_O)d\boldsymbol{y}_i dx_{i,d} \\
&= H[p(\boldsymbol{y}_i,x_{i,d}|\boldsymbol{X}_O)].
\end{aligned}
\tag{25}
$$

For $\textcircled{2}$:

$$
\begin{aligned}
\textcircled{2} &= \int p(\theta,x_{i,d}|\boldsymbol{X}_O)\left[\log p(x_{i,d}|\theta,\boldsymbol{X}_O) + \int p(\boldsymbol{y}_i|\theta,x_{i,d},\boldsymbol{X}_O)\log p(\boldsymbol{y}_i|\theta,x_{i,d},\boldsymbol{X}_O)d\boldsymbol{y}_i\right]d\theta dx_{i,d} \\
&= \int p(\theta,x_{i,d}|\boldsymbol{X}_O)\left[\int p(\boldsymbol{y}_i|\theta,x_{i,d},\boldsymbol{X}_O)\log p(\boldsymbol{y}_i,x_{i,d}|\theta,\boldsymbol{X}_O)d\boldsymbol{y}_i\right]d\theta dx_{i,d} \\
&= \int p(\boldsymbol{y}_i,x_{i,d},\theta|\boldsymbol{X}_O)\log p(\boldsymbol{y}_i,x_{i,d}|\theta,\boldsymbol{X}_O)d\boldsymbol{y}_i dx_{i,d}d\theta \\
&= -\mathbb{E}_{p(\theta|\boldsymbol{X}_O)}[H[p(\boldsymbol{y}_i,x_{i,d}|\theta,\boldsymbol{X}_O)]].
\end{aligned}
\tag{26}
$$

Thus, the Eq.10 with $\alpha = \frac{1}{2}$ is written as

$$
R_C(x_{i,d},\boldsymbol{X}_O) = \frac{1}{2}(H[p(\boldsymbol{y}_i,x_{i,d}|\boldsymbol{X}_O)] - \mathbb{E}_{p(\theta|\boldsymbol{X}_O)}[H[p(\boldsymbol{y}_i,x_{i,d}|\theta,\boldsymbol{X}_O)]]) = \frac{1}{2}I(\theta,\{\boldsymbol{y}_i,x_{i,d}\}|\boldsymbol{X}_O).
\tag{27}
$$

# D  Active Prediction Evaluation Algorithm

---

**Algorithm 3:** Active prediction task evaluation

---

**Result:** Area under information curve AUIC
**input :** $\mathcal{D}_O$,$\mathcal{D}_U$,$\mathcal{M}$,$f(\cdot)$,$\boldsymbol{Y}$
```
/* Initialization */
```
$\mathcal{D}_O = \emptyset$;
AUIC= 0;
**while** $\mathcal{D}_U \neq \emptyset$ **do**
    ```/* Test time acquisition */```
    Compute EDDI reward $R(x_{i,d}, \mathcal{D}_O)$ for $x_{i,d} \in \mathcal{D}_U$ using Eq.22 for each row $i$;
    Select single $x_{i,d} \in \mathcal{D}_U$ into $\mathcal{D}_O$ for each row $i$ ;        ```// Test time acquisition```
    ```/* Test Evaluation */```
    Predict $\tilde{\boldsymbol{Y}}$ =Predict($\mathcal{M}$,$\mathcal{D}_O$);        ```// Prediction```
    Compute $p = f(\boldsymbol{Y})$ ;        ```// Evaluation```
    AUIC+=$p$;        ```// Compute AUIC value```
**end**

---

# E  Training details

In this section, we give details about the experiment setup and the training acquisition.

## E.1  Training time acquisition

We compute the Icebreaker acquisition functions (Eq.10 or Eq.5) for the entire pool set $\boldsymbol{X}_U$. During the selection ,we apply two heuristics. First, instead of picking the top K values, we first normalize their rewards $r_{id}$ with temperature $T$:

$$w_{id} = \frac{\exp(r_{id}/T)}{\sum_{r_{id}} \exp(r_{id}/T)} \tag{28}$$

Then, we sample $x_{i,d}$ according to their weights $w_{id}$. This is a common trick in the active learning techniques to encourage some exploration [2, 58]. When $T \to 0$, this sampling becomes the maximization. The second heuristic is to balance the selected feature number from the observed and new instances. Specifically, assume we need to select $K$ values from the pool, we use the above procedure to select $\frac{K}{2}$ from the rows that have been queried with at least one feature before and other $\frac{K}{2}$ from the rows that are completely new. This is to balance the proportion of exploiting the observed rows and exploring the new ones. For a fair comparison, the second heuristic is applied for all the baselines as well.

## E.2  Training hyperparameters

**UCI.**   We split the whole data set into the training and test sets with proportion $80\%$ and $20\%$. In order to mimic that some features may not be available for query, we manually mask $20\%$ in the training set. For the imputation task, $40\%$ of the data in the test set are masked as the test target and the remaining $60\%$ are reserved as the test input. For the active prediction, we only mask the target variable in the test set as the test target. We also sample $2\%$ of the data instance as the pre-train data as the model has not learned anything in the beginning and the acquisition is the same as random.

We use 5-dimensional latent variable $\boldsymbol{z}$ and the embedding for each feature $\boldsymbol{e}_d$ has 10 dimensions. $h(\cdot)$ is a neural network with 1 hidden layer of 20 units. The aggregation function $g(\cdot)$ is the summation. For the decoder, it has the structure $5 - 100 - 40 - X$, where $X$ is the output dimensions. The data set is normalized with 0 mean and unit variance. We use $\alpha = 1$ and $\alpha = 0.4$ in Eq.10 for the imputation and the active prediction training time acquisition respectively. We use the learning rate 0.003 for the Adam optimizer and $\epsilon^2 = 0.0003$ for the SGHMC step size. We also use $\tau = 0.99$, and $\epsilon\beta = 0.1$ for the SGHMC hyperparameters. The model is trained with 1500 epochs and 100 mini-batch size. The first 750 epochs are used for the SGHMC burn-in and no $\theta$ samples are recorded. At each training time acquisition, the model selects 25 and 50 values from the pool for the active prediction and the imputation respectively.

**MovieLens-1M.**   The *MovieLens-1m* data set contains 1 million ratings for 3000 movies from 6000 users. Each rating is a categorical data ranging from 1 to 5. We follow the same data pre-processing

Figure 8: (Left) The missing proportion of each feature in MIMIC III. (Middle and Right) The norm of the weights. Zero feature weights indicate the corresponding feature is irrelevant to the target according to the model. This figure is directly taken from [36].

procedure as [13] by selecting 1000 movies and 2000 users with the highest number of ratings. We follow the same settings as the UCI imputation with $0.5\%$ data as the pre-train and $20\%$ of the values in the test set as the targets. The model picks 5000 data points from the pool at the training acquisition followed by a model re-initialization to avoid local optima [9].

The latent and feature embedding dimensions are 100 and 50 respectively. The decoder structure is the same as the UCI setting apart from the input and output dimensions. The learning rate and hyperparameters for Adam and SGHMC are the same as the UCI imputation. We train the model using 300 epochs with 100 batch size. Similarly we use half of the total epochs for the SGHMC burn-in. Each training acquisition selects 2000 values from the pool.

**MIMIC III.** The latent and feature embedding dimensions are the same as the UCI active prediction. The decoder structure is changed to $5 - 100 - 100 - 18$, where 18 is the data dimension of MIMIC III. The step size of SGHMC is changed to $\epsilon^2 = 0.0001$. The model is trained for 500 epochs with 100 batch size. The pre-train data set size is $0.5\%$ of the pool data. Each training acquisition selects 50 values from the pool set. The data normalization is the same as the UCI.

### E.2.1 MIMIC III data set statistics

MIMIC III data set after being processed by [11] is extremely imbalanced, where around $88\%$ of the data has label $0$. Thus, training with this data set will result in a lazy model that only outputs label 0. Typically additional pre-processing method for such data set is needed. In this project, we manually balance the data by taking an equal number of instances with label 0 and 1, which forms a new, balanced data set. We do the same for the test set as well. Figure 8 (Left) shows the feature label and its missing proportion in MIMIC III. Table 1 shows the acronym of each label. From Figure 8 (middle and right), we can observe there is a clear shift of importance for *Glucose*. We hypothesize the reason is that the relationship of *Glucose* and target is less linear and cannot be captured by the linear model. For the Icebreaker, when the training data set is small, it is easier to pick up simple relationships. Thus, *Glucose* seems to be less relevant in the beginning. But as the data set grows, Icebreaker can capture non-linear dependencies and start to value the importance of *Glucose*.

## F  Additional UCI Results

For imputation task, we also evaluate the performance of Icebreaker on *Concrete* and *Wine quality* data sets. For the active prediction, we investigate its performance and feature selection strategy in a

| Acronym | Label | Abbrv. | Label |
|---------|-------|--------|-------|
| *Cap.* | Capillary refill rate | *Glu.* | Glucose |
| *Dia.BP* | Diastolic blood pressure | *HR* | Heart Rate |
| *Ins.Oxy* | Fraction inspired oxygen | *Hei.* | Height |
| *GCS:E* | GCS: eye opening | *MBP* | Mean blood pressure |
| *GCS:M* | GCS: motor response | *Oxy.Sat* | Oxygen saturation |
| *GCS:T* | GCS: total | *Res.R* | Respiratory rate |
| *GCS:V* | GCS: verbal response | *Temp* | Temperature |
| *Wei.* | Weight | | |

Table 1: Label and its Acronym

(a) NLL Curve

(b) Long Tail Selection

Figure 9

simple data set called *Energy*. For all these experiments, we follow the experiment setup mentioned previously.

**Imputation.** Figure 9a shows the imputation NLL curve as the data set grows. As expected, Icebreaker outperforms the baselines, especially when the training data size is small. Figure 9b shows the selection pattern of Icebreaker, where we can observe a long-tailed strategy similar to *Boston housing* and *MovieLens-1m* imputation.

**Active Prediction.** Figure 10 shows the AUIC curve as training data set size grows. We can see Icebreaker still outperforms the other 3 baselines by a small margin. The possible reason is that the *Energy* data set is a very simple data set with clear informative variables. We choose this set for the purpose of diagnosing the selection strategy of Icebreaker rather than achieving significant improvement over others. To investigate the strategy of Icebreaker, we group the features in *Energy* data set into 4 groups: Useful for target, Useful, Harder to learn and Useless based on the middle panel in Figure 11.

The x-axis in the middle panel of Figure 11 represents the sorted target value from low to high. Y-axis indicates the feature values corresponding to the target. We can see for the blue line, it has a

Figure 10: AUIC curve w.r.t. *Energy* data set size

clear boundary at target value $-0.3$ and the oscillation of this line is not large compared to others. Thus, it acts as an indicator feature to separate the small and large target values, which is the most useful one for predicting target. As for the red curve, it still has a relatively clear boundary but the large oscillation indicates this is not a robust feature for the prediction. So we refer to it as "Useful". Similar for green curve, its boundary is less clear and it has a even larger oscillation. Therefore, we call it "harder to learn". As for the black curve, it acts as the pure noise and has no clear correlations to target variables. We classify this as "Useless" features.

From the left panel in Figure 11, initially, "Harder to Learn" and "Useless" features are selected the most. This is because the objective Eq.10 encourages the model to find the informative but hard features. Due to the initially scarce training data, the model successfully figures out they are hard to learn but fails to identify which one is more informative. Thus, the selected elements for both features increases in a similar trend. With the data set growing, the model finds out the useless feature. Although it is hard to learn, the model still reduces its query frequency. As for the other two useful features, the model starts to select more of them after 800 data points.

The right panel in Figure 11 shows the initial choice made by the model during the active prediction. There are actually two features that can be classified as 'Useful for target'. But we only plot one of them in the left and middle panel of Figure 11 for clarity. The other one is plotted in the right panel of Figure 11 with light blue. It is the same for 'Useful' features.

It is expected that the 'Useful for target' feature is regarded as the most important one by the model though they are not selected the most in training. 'Useful' and 'Harder to Learn' features are also selected with the number decreasing according to their importance. As expected, the 'Useless' features are not selected at all. Thus, the Icebreaker can indeed discover the important features and select the hard ones among them.

Figure 11: (Left) Accumulated feature number (Middle) Correlations between the features and target (Right) Initial choice at the test time acquisition.