[Reviews · NeurIPS 2019]

Reviewer 1



The primary originality of this paper derives from dealing with active-learning regime with little or no data. This is an extremely important problem for ML, especially as ML is applied to more real-world domains where data is minimal and collection is expensive. The significance of this problem is therefore of high significance. I will discuss the significance their approach to the problem below. Related to this first point, the authors do a fantastic job of situating themselves in the wider active-learning literature, highlighting where there “ice-problem” sits and specifying its unique differences to alternative active learning scenarios. They also motivate this all with real-world examples, which I found particularly helpful and helped with clarity. The model itself is by no means novel, and in that sense, not very significant. The potential significance derives from their inference algorithm - whereby amortized inference is used for local latent variables and sampling methods used for model parameters. The authors make some claims about the significance of this contribution - i.e. allows model to scale with dataset size, while retaining accurate posterior estimates for weights. The discussion of the acquisition functions is extremely clear and well written but the selection method for imputation is not particularly novel as an algorithm - however, it may be in the context of imputation (please see improvements section on related work). The selection method for target prediction is far more interesting and potentially significant. Again, I am unsure if the proposed conditional MI has been suggested before (see improvements on related work), so it is impossible to fully determine the significance for this acquisition function. Nevertheless, a very thorough analysis is given of this acquisition function in terms of exploration (select data to observe new features) and exploitation (select data to make accurate predictions). This was well received - although the final amalgamation of the two different acquisition functions seems rushed - an additional (or re-written) sentence motivating this decision would be warranted. The significance of your work is that you can deal with the ice start problem in active learning. To my understanding, your approach to solving this issue is a) deal with partial observability - however, this is an issue for an element-wise acquisition scheme and b) deal with uncertainty - but this is a problem for active learning system, irrespective of the fact that there may be little or no data. Therefore, it is unclear to me whether your approach adds anything to the ice-start problem per se, as opposed to just element-wise active learning. Second, I would have like to have seen some further discussion about the “novelty” of your inference algorithm. I understand some of your stated motivations - i.e. amortization allows latent inference to scale, while ensuring posterior estimates of parameters are accurate. By why do you prioritize posterior estimates over parameters. One possibility, that you state, is that the SG-HMC algorithm retains the same computational cost as training with a standard VAE. This is obviously important. But some discussion on the importance of maintaining accurate estimates of parameters in terms of active learning would have been warranted. I.e. do you expect this to aid the acquisition functions (whose goal is to reduce posterior entropy over parameters)? The method for embedding (to transform partial observations) is not entirely clear. This could be because I am unfamiliar with the relevant literature. It is not a huge issue, as you do not claim it to be a novel contribution, but an additional sentence would have helped me understand the algorithm as a whole. Second, I find step (i) on line 163 ambiguous. What is x_io when conditioning the posterior over latent variables. I am fairly sure I know what this is, but it took some thinking, going back through the paper, etc. Please clarify this. The equation following line 136 (this equation is not numbered - I feel as though it should be) - why is the complexity (KL[q(z|x)||p(z)]) inside expectation under posterior estimates of parameters? The parameters theta only feature in the accuracy term. Is there a reason for this? I think the related work section needs significant re-writing. There is no mention of previous work on BALD or similar acquisition functions, which are the same as your first function. Moreover, there is no discussion of your proposed novel contributions. In my opinion, this should be worked again from the ground up, as I found it mostly uninformative. Very small point, grammar error on line 77.

Reviewer 2



This paper addresses the ice-start problem where the costs of collecting training data and the costs of collecting element-wise features are both considered. The authors propose a Bayesian deep latent gaussian model (BELGAM) that allows to quantify the uncertainty on weights, and derive a novel inference method. Experiments are performed on two tasks, imputation and active prediction, to demonstrate the effectiveness of the proposed model. This paper is not clearly written and can be better structured. It aims to solve two test tasks (1) imputation and (2) active prediction, however, these two tasks are not clearly defined. I have been working on active learning problems, but I still found it is difficult to follow this paper. I would suggest the authors to focus on addressing one of the two tasks. For example, add a new section of problem definition that precisely defines the objective of the task (move some details from Appendix), and then present how the proposed model can be used to well solve this problem. Some acronyms are used before they are defined in the paper, for example, what is NLL? What is AUIC curve?

Reviewer 3



The introduction is very well written and the method and problem setting are well motivated. The methods are not all original, but are rather an interesting use-case/application of existing methods and are a useful contribution. In section 2, the authors present the BELGAM model, but should cite similar models (ie BNNs in the context of a VAE) that have been used for LVMs in the past (for example, Johnson et al 2016) Section 2 introduces a lot of notation, which could be better summarized in a table. I couldn’t quite follow the explanation of the PVAE; is S_i a matrix or a vector? How do you compute KL(q(theta | X)||p(theta))? Since q(theta|X) is only sampled via a Markov chain, how do you compute its entropy and therefore take its gradient? The experimental validation is quite extensive and is a strength of the paper. Johnson, Matthew, et al. "Composing graphical models with neural networks for structured representations and fast inference." Advances in neural information processing systems. 2016.

[Author Response · NeurIPS 2019]

We thank all the reviewers for their time to provide comments and the positive feedback from reviewer 1 and 4. Also, we appreciate all reviewers for their correct summarisation of our contributions as (1) Introduce the ice-start problem motivated by real-world examples and proposed a method to tackle it. (2) Propose a novel inference algorithm by combining amortized inference of local latent variables and a sampling approach for global weights. (3) Propose novel element-wise acquisition objectives. In the following, we will address the questions proposed by each reviewer.

> Clarify definition on two test problems (Reviewer 3) Thanks for the suggestion. We will focus on the imputation task and make the task setting more clear in the beginning of section 2 and present the prediction task as an extension. We believe that it is important to include two test tasks as these two tasks together cover the most typical real-world problems. This demonstrates the broad applicability of the proposed method. Note that we have indeed included the detailed procedures of both tasks in appendix B.2 Algorithm 3 (imputation) and 4 (active prediction). In the revised version, we will also add a problem definition section for these two tasks in main text.

> Acronyms (Reviewer 3) We will add the NLL acronym in line 80. For AUIC, we used its full name in line 205. The details about AUIC are included in algorithm 4 in appendix B.2. In the revised version, we will move it to the problem definition section in the main text.

> BELGAM related work (Reviewer 4) Thanks for pointing to further related work, which will be included in revised version. However, we would like to highlight our novel inference method for BELGAM. The previous related methods pose strong restrictions/assumptions of the model. For example, they typically require the forward model to be conjugated to the prior or use mean-field approximations for the posterior. Our proposed method uses a sampling approach, which is theoretically guaranteed to give the optimal posterior under mild conditions. In BELGAM section, we will use simple examples to help explaining the generative model for better contextualization.

> Questions about the computation of KL term. (Reviewer 4) The KL term $\mathrm{KL}(q(\theta|\boldsymbol{X})||p(\theta))$ is indeed intractable due to MCMC sampling. However, this term appears in the line 136 and is only used to train the encoder parameter $\phi$. Thus, the gradient is taken w.r.t. $\phi$, and the gradient contribution of this term is zero.

> Encoder for partial observations (Reviewer 1, 4) In the main text, we only briefly mentioned its structure because it is not our novel contribution. We will add a detailed introduction of this encoder in the appendix of the revised version. The term $x_{io}$ represents the observed entries for row $i$. $S_i$ is indeed a matrix. Because for each row $i$, we have some observed features. Each feature has a vector embedding. We concatenate each observed feature value (scalar) with its embedding (vector) to form a vectorised representation of this specific feature. Then, we group all observed features for row $i$ and form its matrix representation $S_i$.

> Notation (Reviewer 4) We will add a table that summarizes the notations used in this paper in the appendix for clarity.

> Difference to basic element-wise AL and related work section (Reviewer 1) We will extend and include more details on the BALD objective in the "Data-wise active learning" subsection, and the comparison to the traditional element-wise active learning (element-wise AL) in the "Feature-wise active learning" subsection in related work. Briefly, we agree that ice-start problem is tightly related to the high-level idea of element-wise AL. However we argue that existing work of element-wise AL methods are commonly restricted to a particular application, such as classification, where a fully-observed test inputs are required; or associated with strong assumptions, such as linear missing data model (e.g. matrix completion) or heuristic acquisition objectives. Thus, they cannot handle the problems like active prediction (test task also has query budget) or imputation tasks with highly non-linear relationship in real-life applications.

>Inference algorithm description (Reviewer 1) We thank reviewer 1 for appreciating our contribution of this novel inference algorithm. We will extend this part, add relevant citations and distinctions compared to previous work in the revised version. Briefly, there is little previous work that uses Bayesian treatment over weight parameters in the context of VAE. The previous inference algorithm either relies on the conjugacy of forward and prior distribution, or strong assumptions over posterior approximations e.g. mean-field. Our proposed method does not requires strong assumptions over posterior, and is guaranteed to give accurate posterior samples under mild conditions. The better posterior estimates not only help with the prediction accuracy but will also indeed aid the acquisition because the acquisition involves the expectation over the posterior. Thus, an inaccurate posterior can result in a poor approximation of the acquisition and lead to the query of the uninformative feature.

> Questions about KL term inside expectation. (Reviewer 1) The reason for writing $\mathrm{KL}(q(\boldsymbol{z}_i|\boldsymbol{x}_i)||p(\boldsymbol{z}_i))$ inside the expectation is because we factorize the posterior in line 126 for computational efficiency. Thus, this factorization decouples the dependency of local latent variables with global weights. In addition, the local variable $\boldsymbol{Z}$ is amortized through each individual $\boldsymbol{x}_i$. The $\mathrm{KL}(q(\boldsymbol{Z}|\boldsymbol{X})||p(\boldsymbol{Z}))$ can be simply written as a summation of $\mathrm{KL}(q(\boldsymbol{z}_i|\boldsymbol{x}_i)|p(\boldsymbol{z}_i)))$ for all $\boldsymbol{x}_i$. Thus, putting it inside the expectation does not affect its value and we can write it outside in the revised version.

> Novelty and combination of acquisition functions (Reviewer 1) To the best of our knowledge, the conditional mutual information is novel in the context of element-wise AL and the previous work is more based on heuristics (e.g. feature variance and model improvement). The combination is based on our intuition as described in lines 166-168. But it indeed corresponds to an information-theoretic quantity (discussed in the paper line 192). This gives an additional warranty for its validity.

[Meta-Review · NeurIPS 2019]

The paper proposes an interesting strategy for active learning in which the model must be learned from little or no data. This is a useful problem to study in practice (which the authors provide examples). As reviewers 2 and 4 note, I recommend polishing the related work particularly in the connections for the BELGAM model. Even the original DLGM paper of Rezende et al. (2014) use priors for the network parameters; but they do MAP estimation.